# Assessment of Outdoor Air Temperature with Different Shaded Area within an Urban University Campus in Hot-Humid Climate

**Sheikh Ahmad Zaki [1],\*, Siti Wan Syahidah [1], Mohd Fairuz Shahidan [2], Mardiana Idayu Ahmad [3],\*, Fitri Yakub [1], Mohamad Zaki Hassan [4] and Mohd Yusof Md Daud [4]**

1   Malaysia-Japan International Institute of Technology, Universiti Teknologi Malaysia, Kuala Lumpur 54100, Malaysia; sitiwansyahidah@gmail.com (S.W.S.); mfitri.kl@utm.my (F.Y.)
2   Faculty of Design and Architecture, Universiti Putra Malaysia, Serdang 43400, Malaysia; mohdfairuz@upm.edu.my
3   Environmental Technology Division, School of Industrial Technology, Universiti Sains Malaysia, Penang 11800, Malaysia
4   Razak Faculty of Technology and Informatics, Universiti Teknologi Malaysia, Jalan Semarak, Kuala Lumpur 54100, Malaysia; mzaki.kl@utm.my (M.Z.H.); yusof.kl@utm.my (M.Y.M.D.)
\*   Correspondence: sheikh.kl@utm.my (S.A.Z.); mardianaidayu@usm.my (M.I.A.)

**Abstract:** This study investigated the variation of outdoor air temperature in the shaded area covered by buildings in an urban university campus in Malaysia. In-situ field measurements were conducted to measure the distribution of outdoor air temperature at eight different locations for seven days. Meanwhile, the building-induced shadows were generated using the AutoCAD Revit software to investigate the air temperature change. The study used four urban morphological parameters namely building to greenery ratio, sky view factor (SVF), and height-to-street width (*H/W*) ratio. The relationship between building-induced shadow and outdoor air temperature ($T_{out}$) obtained from the in-situ measurement was investigated. The results showed that the building-induced shadows could lower air temperature. It can be noted that a high ratio of building to greenery resulted in a higher air temperature. In contrast, the area with a low SVF value due to the combination of prolonged shading by buildings and trees had a lower air temperature. Thus, the area with a high building ratio, low greenery ratio, higher SVF value, and low *H/W* ratio potentially has a higher outdoor air temperature. Conclusively, combination of building shading created by appropriate ratio of building morphology and sufficient greenery able to improve the microclimate of a campus area.

**Keywords:** microclimate; building morphology; university campus; building-induced shadow; field measurement

## 1. Introduction

Although urbanization uplifts the technological livelihood of urban inhabitants, the process has significant impacts on urban environment and urban climate. For instance, the development of residential, industrial and commercial areas issues changes in the natural landscape of dense vegetation to concrete, impervious roughness structures of urban morphology [1]. Changing urban morphology has contributed to urban climate change (e.g., increasing urban temperatures, reducing urban ventilation). Heat accumulation, heat trapping, increased roughness, and reduced evaporation are among the explanations for the climate change caused by urbanization [2]. Besides, long-wave and

solar radiation interactions are affected by shadowing, reflections between buildings, and diminished sky-view factors.

A local urban microclimate has an impact on a building's energy demand and human comfort and health in urban areas [3]. There is a growing number of studies on the impact of urban morphological parameters on microclimate. To date, growing attention is given to the investigation of the relationship between urban morphological parameters and microclimate characteristics [2–6]. Thus, this paper introduces and discusses the designated urban morphological parameters correlated with the urban microclimate characteristics investigated throughout the study. The parameters include building height characteristics, height-to-street width (*H/W*) ratio, and built-up percentage.

Existing research recognizes that building height plays a critical role in urban microclimate. Burian et al. [7] investigated building height characteristics in various areas of different land uses and indicated that the classification of building heights based on the land use purpose helps to determine the temperature and air ventilation conditions in that particular area. Several recent studies of urban microclimate based on the building height characteristics were carried out to investigate the thermal exchange in the zones located between buildings [8–11]. Thus, some researchers classified the building height distributions in regard to the land use purpose; however, for an area with a specific land use purpose such as a university campus, building heights are classified into low, medium, and high-rise [12].

The heterogeneity of building heights creates different thermal exchanges in regard to solar radiation and wind flow, both of which influence air and surface temperatures. Investigation of the low thermal performance of a high-rise and high-density residential area found that the area with a high canyon geometry played an effective role of dissipating a high radiant energy surplus (70–80%) during the daytime through turbulent transfer and releasing the remaining stored radiant energy surplus (30–20%) at nighttime [13,14].

Furthermore, building height is shown to be crucial in urban design to reduce solar radiation effect [15–18]. Essentially, urban building height should restrict the amount of solar radiation from reaching the pedestrian level in order to lower the temperature in a street canyon and improve the urban microclimate. Boubia and Awbi [19] examined that the variation of street surfaces and ground shading effects due to building height and street width resulted in the difference of surface and air temperature distributions. The study identified that temperature is highly correlated to street geometry and sky view factor (SVF). Therefore, building height is an important element in improving urban microclimate at the pedestrian level as it influences the amount of radiation through building shading and affects natural ventilation.

According to Burian et al. [7], the *H/W* ratio can be defined as the ratio of the average building height to the horizontal distance (or street width) between two adjacent buildings. The *H/W* ratio is also known as 'street aspect ratio' [20,21].

In a tropical urban environment, the combination effect of high temperature and prolonged sunlight exposure in the daytime leads to serious issues of heat anxiety resulting in heat-related sickness and outdoor thermal inconvenience among urban occupants [22]. The *H/W* ratio is important in determining the inbound short wave radiation and leaving long wave radiation, as well as mitigating the effect of UHI. More recently, various findings concerning the *H/W* ratio have been reported in literature. In Singapore, Goh and Chang [15] demonstrated a positive statistical correlation between the *H/W* ratio and the nighttime heat island which caused a slight increase in the temperature. Bourbia and Awbi [19] evaluated the effects of two different urban street canyons distinguished by the *H/W* ratio in the hot and arid climate indicating that the lower *H/W* ratio resulted in higher air temperature due to high exposure to solar radiation.

The influence of *H/W* on urban microclimate can lead to the modification of air and surface temperatures [23]. The *H/W* ratio is highly correlated with the nighttime temperature, whereby a higher *H/W* ratio leads to a greater nighttime air temperature and a lower daytime temperature [17,24,25]. In this regard, building height and canyon width, which are the two parameters to determine the *H/W*

ratio, affect the reflection of solar radiation during the daytime and the dispersion of emitted heat at nighttime, thus contributing to the diurnal and nocturnal temperature variations. Cities located in the hot and humid climate experience a prolonged exposure to solar radiation where local air temperature is always high. However, the decrease of local air temperature is possible with the aid of shading factors such as buildings or vegetation [26]. The shading effect can be determined by the *H/W* ratio, which can be essentially useful for UHI mitigation strategies in the cities with hot and humid climate.

According to Watson and Johnson [27], the term sky view factor (SVF) is defined as the proportion of the radiation obtained by the horizontal surface to that emitted by the hemispheric environment. SVF is frequently identified to quantify sky openness and describe irradiative properties; the SVF value of an area ranges between zero and one. In a typical open space, the SVF value is one, but when there are obstacles i.e., buildings or trees that impede the sky view, the SVF value decreases to zero [28]. Furthermore, the SVF of an urban canyon geometry is essentially correlated with radiation, air [29], and the humidity–water estimation that explains the heat island development. In this regard, a low SVF reduces the UHI effect by reducing the nighttime temperature and increasing the diurnal shading effect [13,30]. For this reason, SVF presents significant and good correlation with outdoor air temperature during the daytime. At nighttime, the correlation is particularly weak indicating that a net long wave loss during the nighttime is not only caused by SVF, but also other reasons such as surface material.

Oke [31] stated that besides building geometry, the planting of street trees reduces the value of SVF and consequently decreases the radiative heat loss originated from trapped heat in buildings and streets. In another study, Shashua-bar and Hoffman [32] discussed that a fractional unshaded area of tree canopy reduces the SVF. To date, SVF has become the main parameter of interest among researchers especially in cities characterized by the hot and humid climate [33,34]. Such studies are crucial to propose and implement mitigation strategies of the UHI effect and increase the thermal comfort level among urban inhabitants. Therefore, the parameters including building height, height-to-street width (*H/W*) ratio, and built-up percentage are essential to be correlated with the effects on microclimate.

To address the aforementioned issue, this study aims to investigate the variation of outdoor air temperature distributions at shaded areas covered by buildings in an urban university campus located in Kuala Lumpur, Malaysia, based on the in-situ field measurement. This study is focused on the building shading effect on the shaded areas covered and affected by building-induced shadows.

## 2. Methods

### 2.1. Field Measurement

In Malaysia, the climate of Kuala Lumpur is influenced by the Titiwangsa Mountains to the east and the island of Sumatra, Indonesia to the west. Air temperature is usually constant throughout the year with monthly average lows ranged between 23.4 to −24.6 °C and never drop below 14.4 °C, while monthly average highs are from 32.0 to −33.0 °C and never exceed 38.5 °C [35].

The field measurement was carried out at Universiti Teknologi Malaysia Kuala Lumpur (UTMKL), Malaysia from 8–14 April 2015. The outdoor air temperature data were collected at eight measurement points labelled as A, B, C, D, E, F, G, and H as shown in Figure 1. The Onset HOBO (U12-013) data logger was equipped with the built-in air temperature and relative humidity sensors with the accuracies of ±0.35 °C and ±2.5%, respectively, as shown in Figure 2. The instrument was used for the in-situ data measurement and covered with a shield to prevent direct exposure to solar radiation that can affect the climatic data collection. The instrument setup details can be referred in Zaki et al. [33]. Table 1 displays the description of each measurement location. Two weather stations were used to monitor air temperatures from 8–14 April 2015. The first station (WS1) was setup at the rooftop of the second highest building located at the centre of this campus. The building height was about 53.0 m and the sensors were installed at about 15 m from the rooftop. Another weather station (WS3) was installed at

about 2.5 m from ground level. This station located near to main gate of this campus and in front of the highest building.

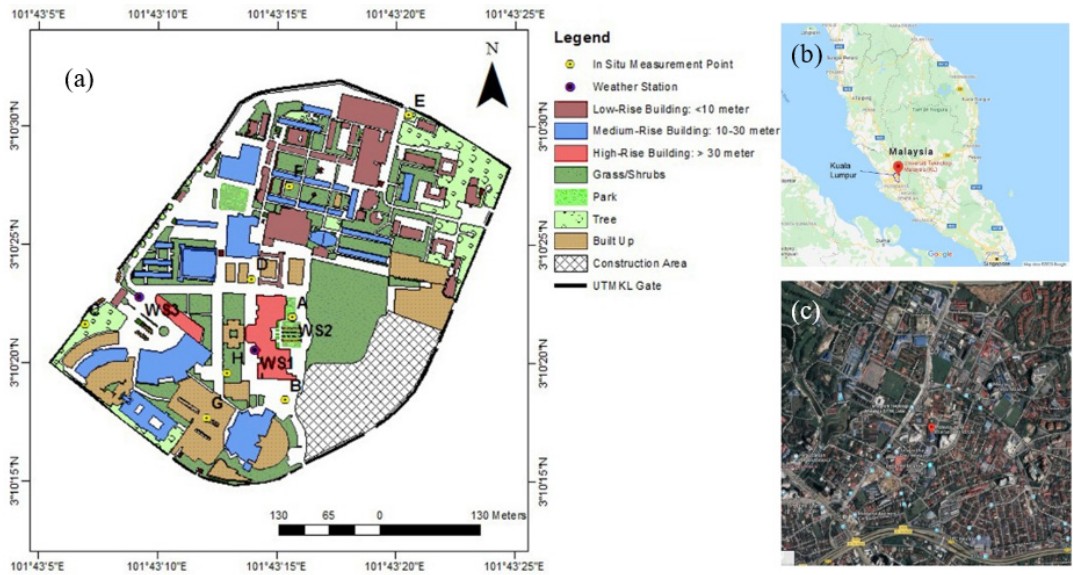

**Figure 1.** (**a**) The map of field measurement points labelled as A, B, C, D, E, F, G, and H. The descriptors are shown in Table 1. WS1, WS2, and WS3 refer to the weather station 1, weather station 2, and weather station 3, respectively; (**b**) Map of Malaysia (Source: Google Maps, Country of Malaysia, 25th June 2020); (**c**) Map of the investigated campus with surrounding areas (Source: Google Satellite, 25th June 2020).

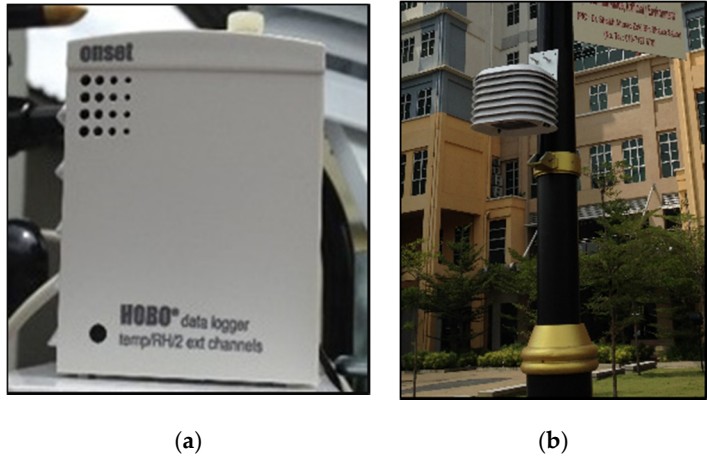

(**a**)  (**b**)

**Figure 2.** Instruments used for the in-situ data collection (**a**) Onset Hobo data logger (**b**) Set-up of the Onset Hobo data logger with a solar radiation shield.

**Table 1.** Descriptions of the in-situ measurement locations.

| Point | Landmark Location | Point | Landmark Location |
|:---:|:---:|:---:|:---:|
| A | UTMKL field | E | Noise and Vibration Lab |
| B | Construction area | F | Zone between Block A and Block B |
| C | UTMKL main gate | G | Parking lot |
| D | Tennis court | H | Zone between Razak Tower and MJIIT |

Note: UTMKL—Universiti Teknologi Malaysia Kuala Lumpur; MJIIT—Malaysia-Japan International Institute of Technology.

## 2.2. Building-Induced Shadows

This study only focused on the air temperature distributions at building-shaded areas or building shadows instead of vegetation shadows since most of the shaded areas in the UTMKL campus were covered and affected by building-induced shadows. The image of the building-induced shadow in UTMKL was produced using the AutoCAD Revit software (Autodesk, Mill Valley, CA, USA). The base map of UTMKL obtained from the Geographic Information System (GIS) database was exported to the AutoCAD Revit software to generate building shadows with similar scales and coordinates used in the GIS database. The items exported were stored into different layers such as building layer, greenery layer, and construction area layer. The features in the building layer were extruded according to building height obtained from the field measurement to create three-dimensional buildings of UTMKL.

The three-dimensional buildings of UTMKL were the main data source to create the building shadows. The building-induced shadows were created by setting the apparent motion of the sun. The sun orientation setting involved the setting of the solar studies, location, date, time, and time interval. The location was set to Kuala Lumpur city, and the date and time were set to the dates of the in-situ measurement, from 8–14 April 2015. The interval setting of one hour was used to observe the hourly building shadows from sun rise to sun set within the period of the in-situ measurement. Based on the setting, the building shadows were generated from the sun rise to the sun set of the day.

In order to check the validity of the shadows generated in the AutoCAD Revit software, a video of building-induced shadows was recorded between the MJIIT building and the UTMKL Razak tower on 13th February 2014. The video was recorded using Recolo Interval Camera.

The shadows from the video and AutoCAD Revit are shown in Figure 3. The findings show that the shadows from both sources change consistently with time.

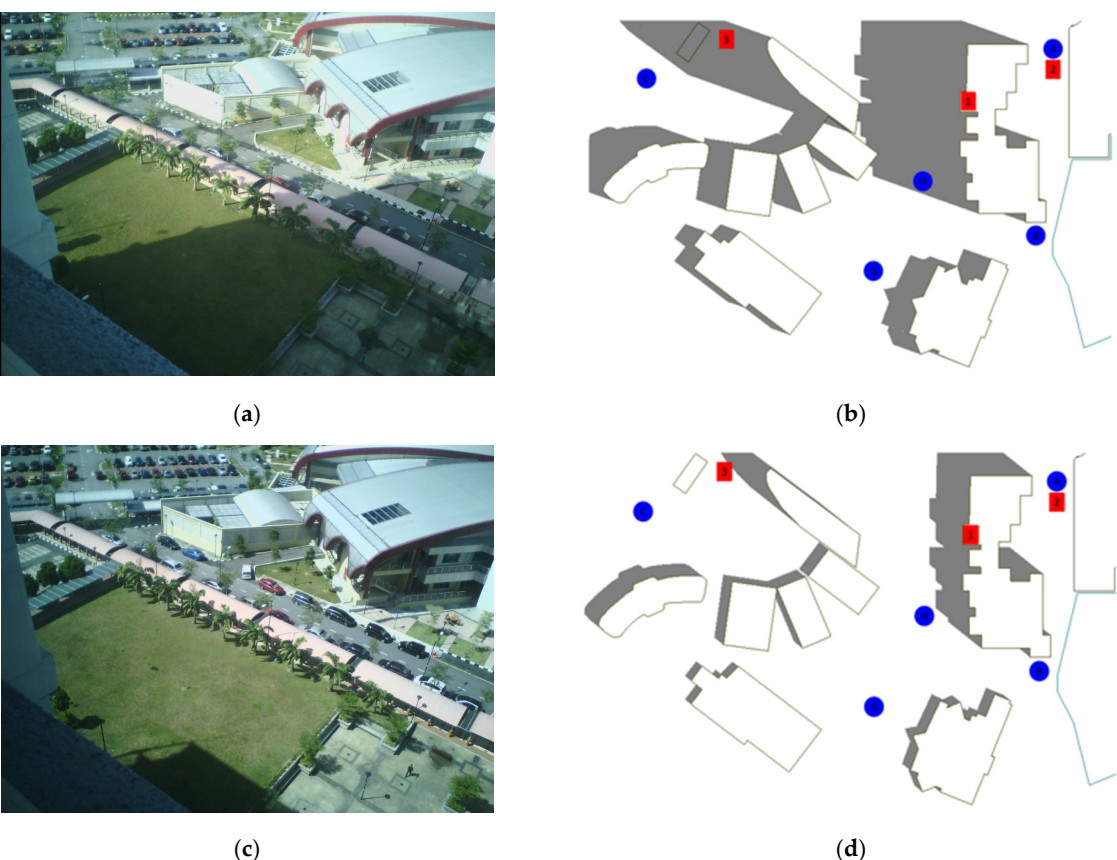

(a)　　　　(b)

(c)　　　　(d)

**Figure 3.** Building-induced shadows from the video captured and from AutoCAD Revit. (**a**) 10:00; (**b**) 10:00; (**c**) 12:00; (**d**) 12:00.

## 2.3. Building Height Characteristics

The UTMKL campus comprised of 78 buildings with heights ranged from 3–83 m (Figure 4). 12 buildings in this campus (15% of the total buildings) stand at 4 m. The building height average of UTMKL was 9.96 m with a standard deviation of 10.77.

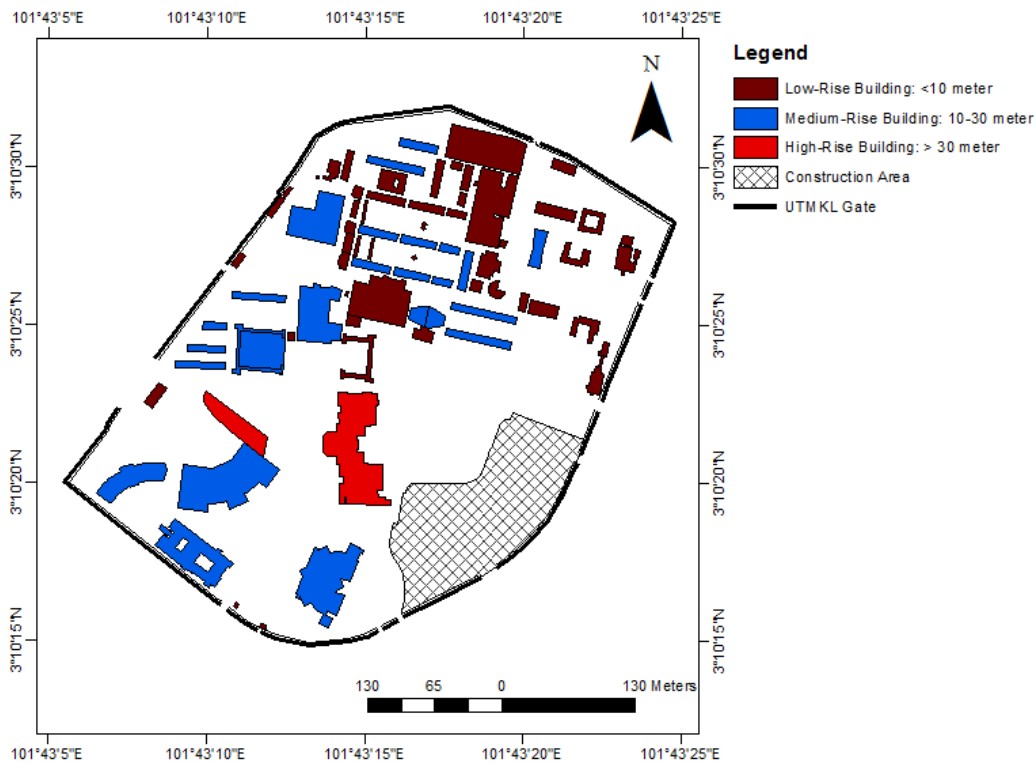

**Figure 4.** Buildings of UTMKL color coded by height classifications i.e., low-rise, medium-rise, and high-rise buildings.

The GIS application was used to map the building height based on low-rise building, medium-rise building, and high-rise building as shown in Figure 4.

## 2.4. Building H/W Ratio

There were 19 locations in UTMKL whose *H/W* ratios were calculated. The locations were selected based on the building height and the canyon formed between the surrounding buildings.

The canyons at 19 locations were between two of similar building types (i.e., low-rise building, medium-rise building, and high-rise building) and the combinations of low-rise and medium-rise, low-rise and high-rise, and medium-rise and high-rise. Most of the low-rise building canyons in UTMKL showed low *H/W* ratio (63.2%). The medium *H/W* ratio (21.1%) were found at the wide building width and between the medium rise buildings with the combinations of buildings from different height categories. High *H/W* ratio (15.8%) were spotted at the zone between the combinations of high-rise buildings with the medium rise and low rise height.

Furthermore, the *H/W* ratios at the field measurement points are provided in Table 2. The highest *H/W* ratio of 1.26 was at in-situ point H. Point H was surrounded by two high-rise buildings, Razak tower (83 m) and MJIIT building (53 m), while the width between the two buildings were 54.4 m. In contrast, in-situ points D and G had low *H/W* ratios of 0.16 and 0.18, respectively due to their locations at open spaces i.e., near the tennis court and at the parking lot, respectively. Moreover, in-situ point D was surrounded by low-rise buildings, while in-situ point G was surrounded by medium-rise buildings. Moreover, the ratio of building to greenery (RBG) in the investigated area as shown in Table 2 were calculated.

**Table 2.** *H/W* ratios at field measurements in UTMKL.

| Location | Photograph of the Real Canyon | *H/W* Cross Section Illustration | *H/W* | Ratio of Building to Greenery (RBG) |
|---|---|---|---|---|
| A: UTMKL field |  |  | 0.49 | 1:1.42 |
| B: Construction area |  |  | 0.89 | 1:0.49 |
| C: UTMKL main gate |  |  | 0.27 | 1:4.32 |
| D: Tennis court |  |  | 0.16 | 1:0.34 |
| E: Noise and Vibration Lab |  |  | 0.20 | 1:1.15 |
| F: Zone between Block A and Block B |  |  | 0.43 | 1:1.01 |
| G: Parking lot |  |  | 0.18 | 1:1.04 |
| H: Zone between Razak Tower and MJIIT |  |  | 1.26 | 1:2.49 |

### 2.5. Sky View Factor (SVF)

The values of SVF at the field measurement points are presented in Table 3. The highest SVF value i.e., 0.97 was at point G which was located at the parking lot area. This indicates that the area was an open space with nearly no obstacles to block the sky view. In addition, a low SVF value of 0.61 was obtained at both points F and C. Point F was located between building blocks, Block A and Block B. In addition, the presence of palm trees at that location led to low SVF. Point C was located near the main gate from which the highest building in UTMKL, the Razak tower was not far. The presence of Razak Tower and trees around the area obstructed the sky view causing low SVF at point C.

The SVF values of the other in-situ measurement points of A, B, D, E, and H were 0.75, 0.79, 0.68, 0.62 and 0.83, respectively. A further discussion on the relationship between SVF and air temperature of the surrounding area is provided in Section 3.3. Since the SVF value also indicates the compactness of an area with buildings, vegetation, or both, it could be used as an additional parameter in town planning or urban development to create a sustainable environment.

**Table 3.** The sky view factor at in-situ point measurements in UTMKL.

| Location | Description | Fish-Eye Photo | SVF |
|:---:|:---:|:---:|:---:|
| A | UTM KL field |  | 0.75 |
| B | Construction area |  | 0.79 |
| C | UTM KL main gate |  | 0.61 |

**Table 3.** *Cont.*

| Location | Description | Fish-Eye Photo | SVF |
|---|---|---|---|
| D | Tennis court |  | 0.68 |
| E | Noise and Vibration Lab |  | 0.62 |
| F | Zone between Block A and Block B |  | 0.61 |
| G | Parking lot |  | 0.97 |
| H | Zone between Razak Tower and MJIIT |  | 0.83 |

## 3. Result and Discussion

### 3.1. Outdoor Air Temperature

The daily average outdoor air temperature, $T_o$ obtained from the in-situ measurements and from two weather stations, WS1 and WS3, are plotted in a graph shown in Figure 5. The highest and lowest daily averages of $T_o$ were observed from 8th April 2015 until 12th April 2015. The highest daily average of $T_o$ from all in-situ measurement points, WS1, and WS3 was observed on 8th April 2015 and the lowest on 12th April 2015. The highest and lowest daily averages of $T_o$ were associated with the weather conditions of that particular day such as rainfall. On 8th April 2015 which was the day with no rainfall data, the highest daily average of $T_o$ was observed, while on 12th April 2015, the daily total

rainfall of 0.8 mm had lowered the daily average of $T_o$. The mean relative humidity and wind speed during the period of measurement were 72% and 1.0 m/s, respectively.

Besides WS1 and WS3, location B shows the highest $T_o$ among other locations. The increased of temperature is due to the intensive exposure of solar radiation. Contrasting to other locations, the low $T_o$ might be due to the effect of building-induced shadow, building $H/W$ ratio, and decreased sky view factor (SVF). The obstruction is mostly due to the building-induced shadow and supported by the existence of random location of vegetation that significantly affects the microclimate of the area.

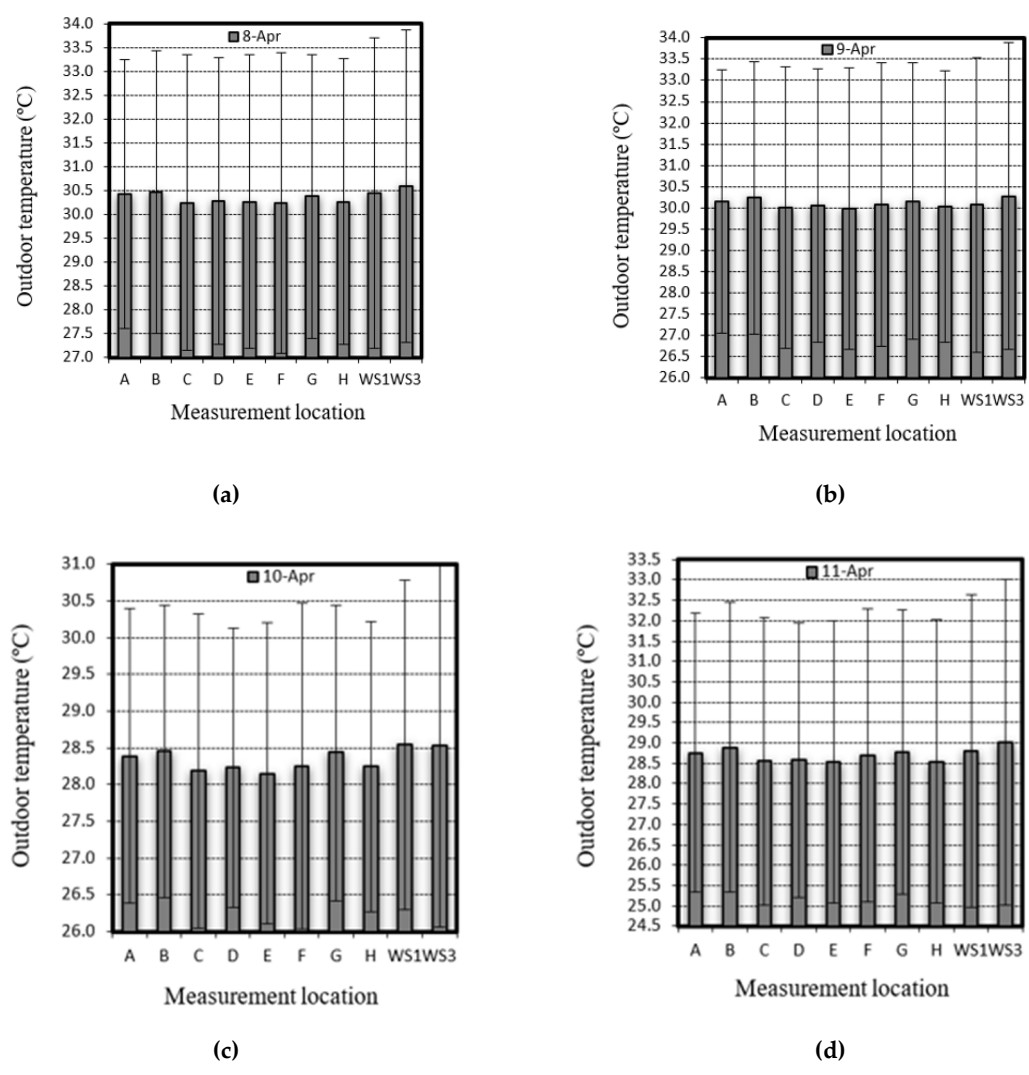

(a)

(b)

(c)

(d)

**Figure 5.** *Cont.*

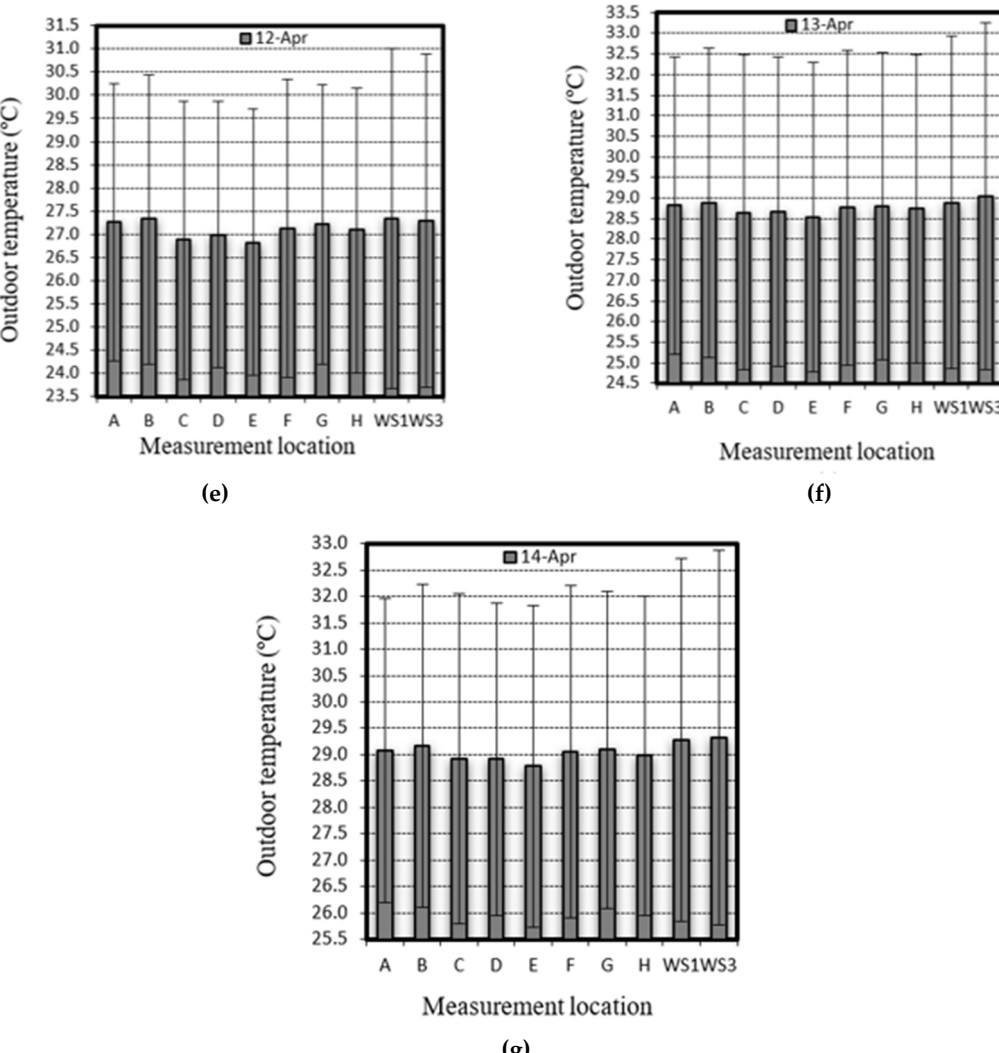

**Figure 5.** Daily average air temperatures of in-situ measurement points (A to H) and weather stations (WS1 and WS3) during the following days of measurement (**a**) 8th April; (**b**) 9th April; (**c**) 10th thApril; (**d**) 11th April; (**e**) 12th April; (**f**) 13th April; (**g**) 14th April. Error bar refers to standard deviation.

## 3.2. Building-Induced Shadows

The building-induced shadows were analysed based on the average daily air temperatures at in-situ measurement points. The daily building-induced shadows observed on each measurement day at 12:00 were compared. Figure 6 presents the daily building-induced shadows at 12:00 from 8th to 14th April 2015. The building-induced shadows at 12:00 were selected due to no occurrence of rain and clear of cloud condition at that time. There are similar building-induced shadows at 12:00 for all-day measurements. Thus, building shadows on 8th April 2015 were used for discussion.

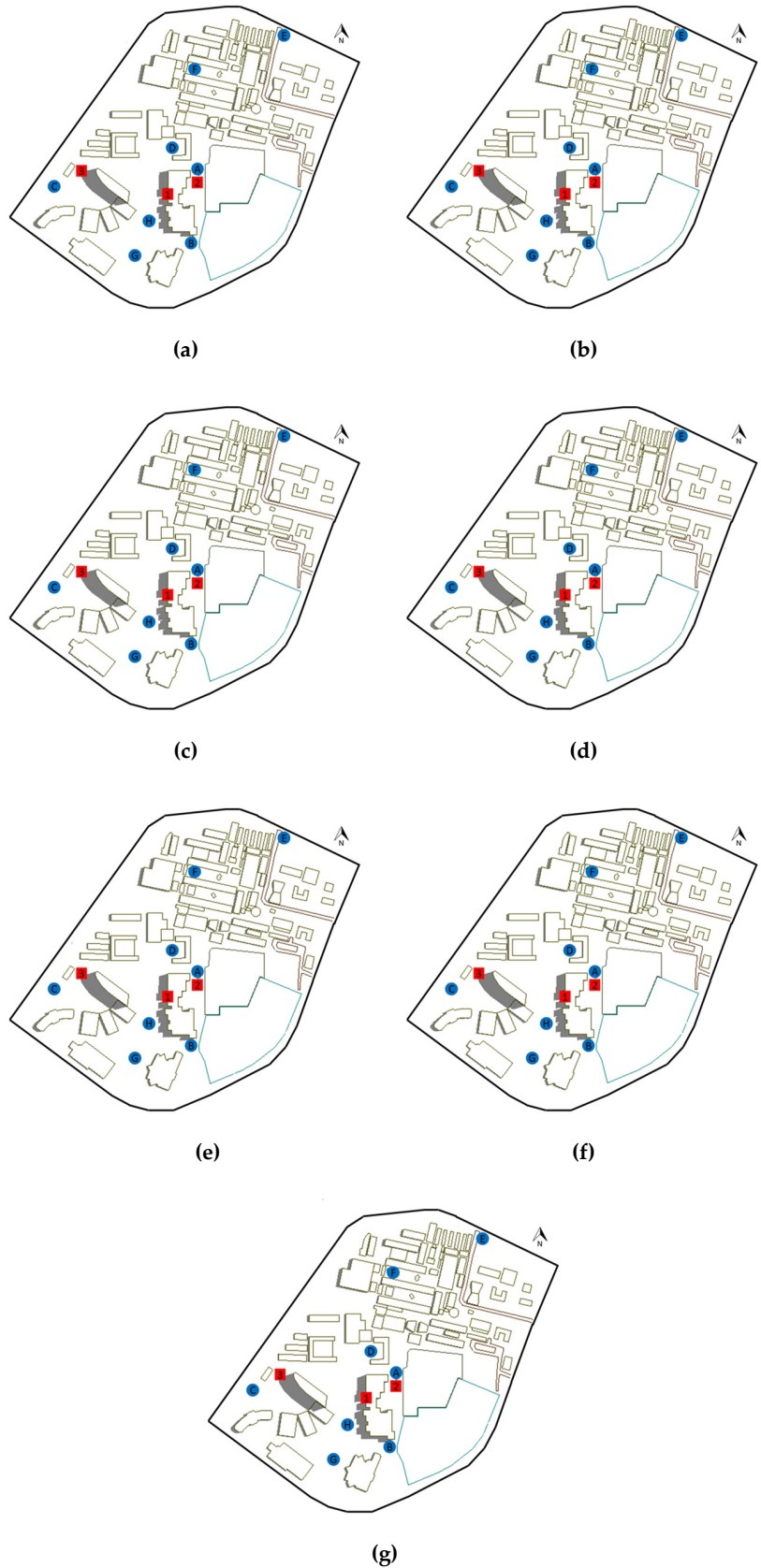

**Figure 6.** Daily building-induced shadows in UTMKL at 12.00 from 8th to 14th April 2015. (**a**) 12:00, 8th April 2015; (**b**) 12:00, 9th April 2015; (**c**) 12:00, 10th April 2015; (**d**) 12:00, 11th April 2015; (**e**) 12:00, 12th April 2015; (**f**) 12:00, 13th April 2015; (**g**) 12:00, 14th April 2015.

Figure 7 shows the building-induced shadows of UTMKL on 8th April 2015. The observation of building-induced shadows was selected on 8th April 2015 to represent the daily building-induced shadows since the building-induced shadows for the seven days were observed to be constant with only very slight changes. The sun rose at 7:10 and set at 19:20. The building-induced shadows lowered the air temperature at the shaded area as shown in Figure 8a. However, point B had a high average air temperature because the point was only covered by building-induced shadows at 19:20 and exposed to the sunshine in the remaining hours as displayed in Figure 8b.

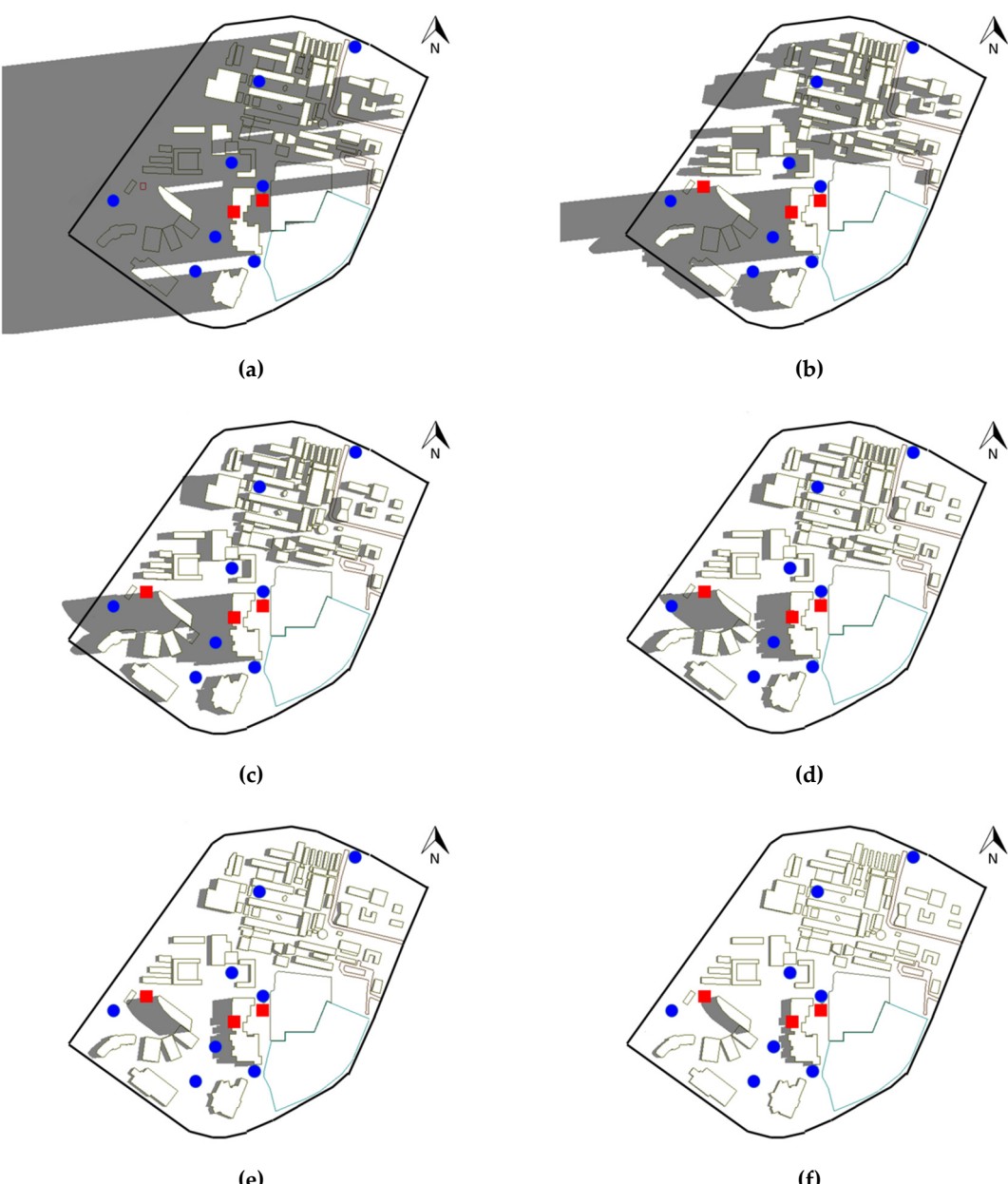

(a)

(b)

(c)

(d)

(e)

(f)

**Figure 7.** *Cont.*

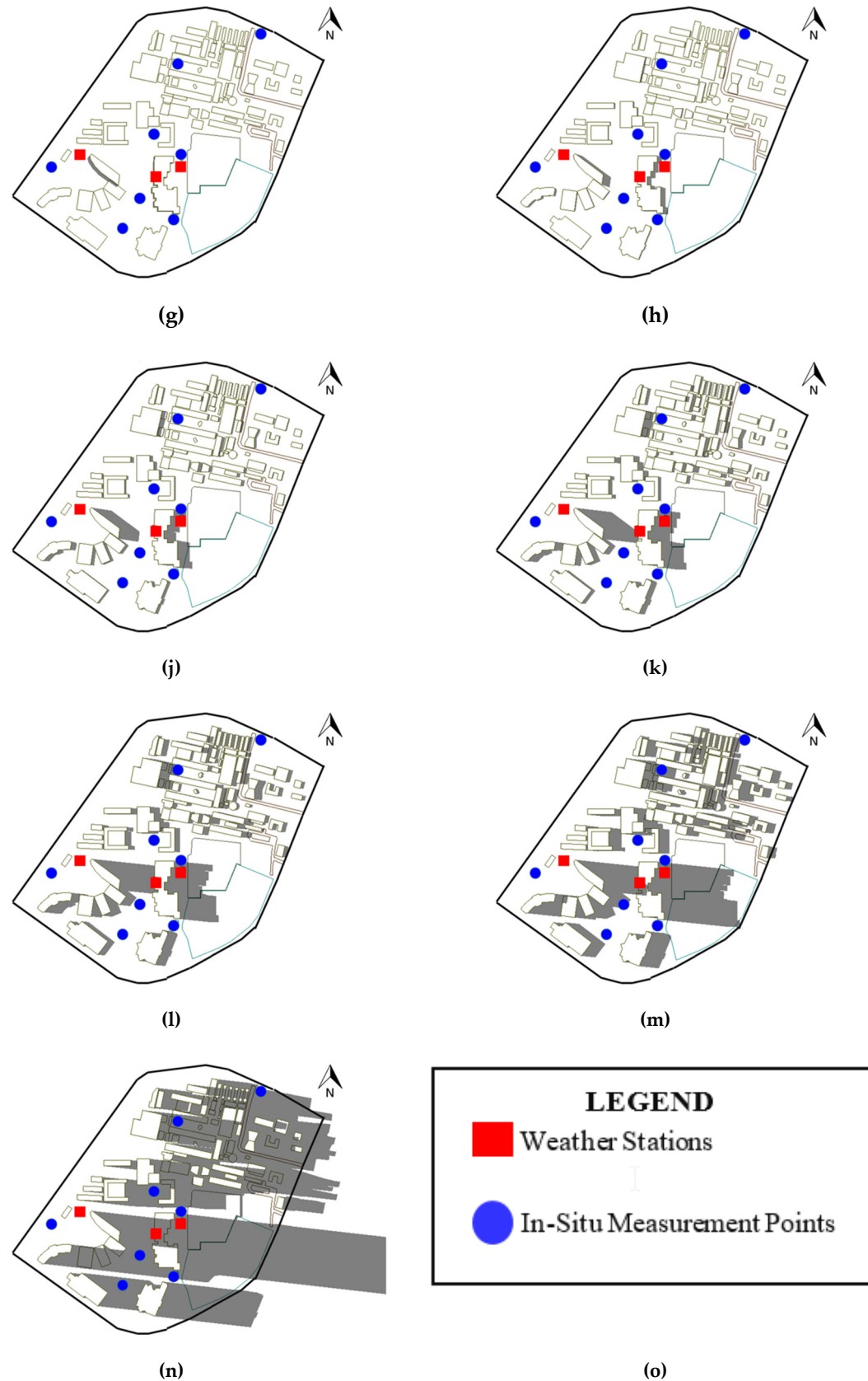

**Figure 7.** Hourly building-induced shadows on 8th April 2015. (**a**) 7:10; (**b**) 8:00; (**c**) 9:00; (**d**) 10:00; (**e**) 11:00; (**f**) 12:00; (**g**) 13:00; (**h**) 14:00; (**j**) 15:00; (**k**) 16:00; (**l**) 17:00; (**m**) 18:00; (**n**) 19:20; (**o**) Legend.

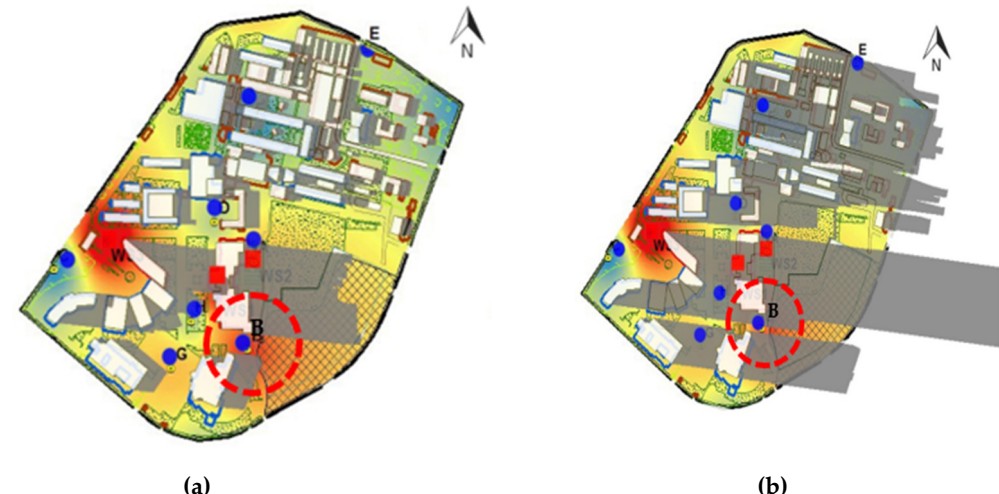

**Figure 8.** Air temperature distributions at the campus area (**a**) at 18:00 (**b**) after 19:00. Grey-shaded areas refer to the building-induced shadows.

The periods of direct exposure to the sunshine (i.e., without building-induced shadow cover) varied by the measurement locations: A (8 h), B (12 h), C (8 h), D (10 h), E (12 h), F (8 h), G (10 h), and H (6 h).

Furthermore, the in-situ $T_o$ measurement data were plotted in a graph based on the hourly average, as shown in Figure 9. Based on the graph, the higher and lower hourly averages of $T_o$ were observed in the daytime and at nighttime, respectively. The daytime and nighttime hours were observed from 7:00 to 18:00 and from 19:00 to 6:00, respectively according to the sun rise and the sun set as studied by Jamei et al. [17]. During the daytime, the hourly average of $T_o$ was highest from 12:00 to 16:00, while at nighttime, the hourly average of $T_o$ was lowest from 0:00 to 6:00.

The highest hourly average of $T_o$ (i.e., 36.7 °C) was observed at 16:00 (8th April 2015), while the lowest hourly average of $T_o$ (i.e., 24.0 °C) was observed at 21:00 (11th April 2015). In general, the hourly average of $T_o$ was primarily influenced by rainfall, where during the highest hourly average of $T_o$, there was no occurrence of rainfall throughout the day. However, during the lowest hourly average of $T_o$, heavy rainfall of 3.2 mm was recorded from 20:00 to 21:00 resulted in the lowest hourly average of $T_{out}$.

Moreover, on 10th April 2015, the hourly average of $T_o$ abruptly dropped at 15:00 due to the occurrence of rainfall with a total of 0.6 mm recorded at the time. Based on Figure 9, the highest $T_o$ was observed at location B. This is due to the absence of building-induced shadow from 8:00 to 18:00. The other locations towards the west denote that $T_o$ is slightly lower in the morning due to the effect of building-induced shadow which starts from 7:00 until 12:00. From noon onwards, the $T_o$ in those areas become higher due to the changes of shadow direction. As the highest $T_o$ was obtained from 12:00 to 16:00, the microclimate at the locations towards the west benefit from the prolonged building-induced shadow effects starting from 14:00 to 19:00. During this period, the $T_o$ of the areas located near to the highest building reduced from 0.5 °C to 1.0 °C depending on the shaded coverage area, *H/W* ratio, and SVF of building and vegetation. The decrease of $T_{out}$ is due to the reduction of long-wave and solar radiation interactions which affects shadowing, reflections between buildings, and diminished SVF [3].

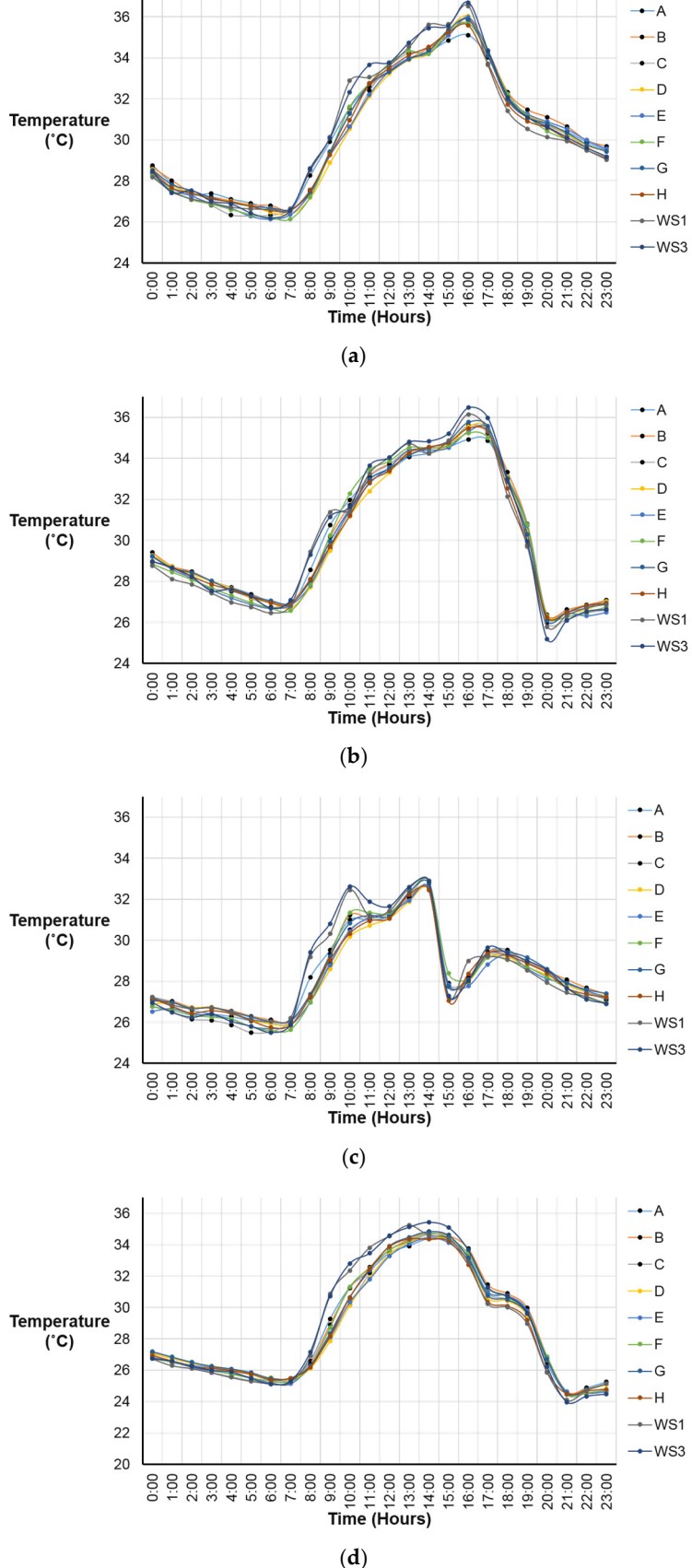

(**a**)

(**b**)

(**c**)

(**d**)

**Figure 9.** *Cont.*

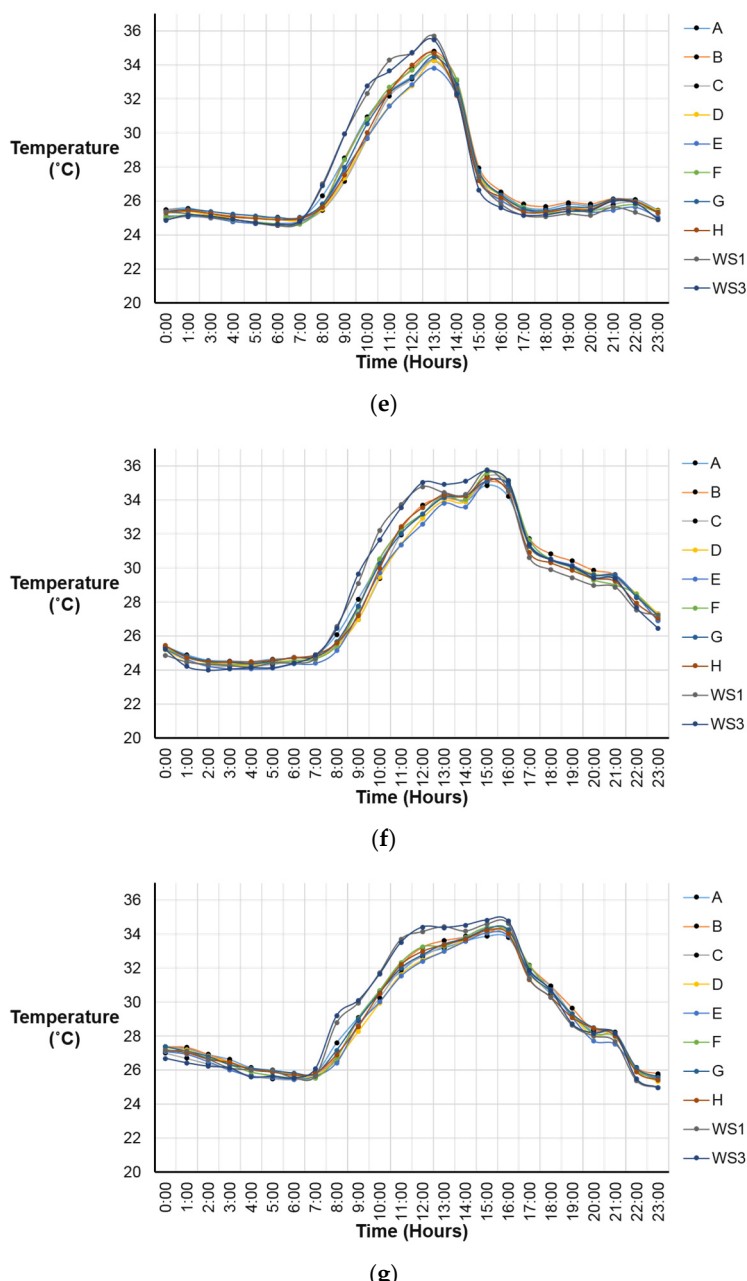

(e)

(f)

(g)

**Figure 9.** Hourly average air temperature at in-situ measurement points on (**a**) 8th April 2015 (**b**) 9th April 2015 (**c**) 10th April 2015 (**d**) 11th April 2015 (**e**) 12th April 2015 (**f**) 13th April 2015 (**g**) 14th April 2015.

*3.3. Relationship between Outdoor Air Temperature and Land Cover*

The land cover features were overlaid with the recorded daily average air temperatures and the detailed analysis within 50 m of the field measurement points, as presented in Table 4. The mapping of UTMKL land cover features and the daily average air temperature interpolation from 8th until 14th April 2015 was displayed in Figure 10.

**Table 4.** Parameters of each measurement location where the area ratio and area percentage are defined as building to greenery (RBG), SVF is the sky view factor, *H/W* is the height-to-width ratio, and the average air temperature within 50 m from the in-situ measurement points.

| Location | Description | RBG | SVF | *H/W* | Average Air Temperature | | | | | | |
|---|---|---|---|---|---|---|---|---|---|---|---|
| | | | | | April (2015) | | | | | | |
| | | | | | 8th | 9th | 10th | 11th | 12th | 13th | 14th |
|  | UTM KL Field | 1:1.42 | 0.75 | 0.49 | 29.0 | 30.2 | 28.4 | 28.8 | 27.3 | 28.8 | 29.1 |
|  | Construction Area | 1:0.49 | 0.79 | 0.89 | 29.1 | 30.2 | 28.5 | 28.9 | 27.3 | 28.9 | 29.2 |
|  | UTM KL Main Gate | 1:4.32 | 0.61 | 0.27 | 28.8 | 30.0 | 28.2 | 28.6 | 26.9 | 28.6 | 28.9 |
|  | Tennis Court | 1:0.34 | 0.68 | 0.16 | 28.8 | 30.0 | 28.2 | 28.6 | 27.0 | 28.7 | 28.9 |
|  | Noise and Vibration Lab | 1:1.15 | 0.62 | 0.20 | 28.7 | 30.0 | 28.2 | 28.5 | 26.8 | 28.5 | 28.8 |
|  | Zone Between Block A and Block B | 1:1.01 | 0.61 | 0.43 | 28.9 | 30.1 | 28.3 | 28.7 | 27.1 | 28.8 | 29.0 |
|  | Parking Lot | 1:1.04 | 0.97 | 0.18 | 28.9 | 30.2 | 28.4 | 28.8 | 27.2 | 28.8 | 29.1 |
|  | Zone between Razak Tower and MJIIT | 1:2.49 | 0.83 | 1.26 | 29.0 | 30.0 | 28.2 | 28.5 | 27.1 | 28.7 | 29.0 |

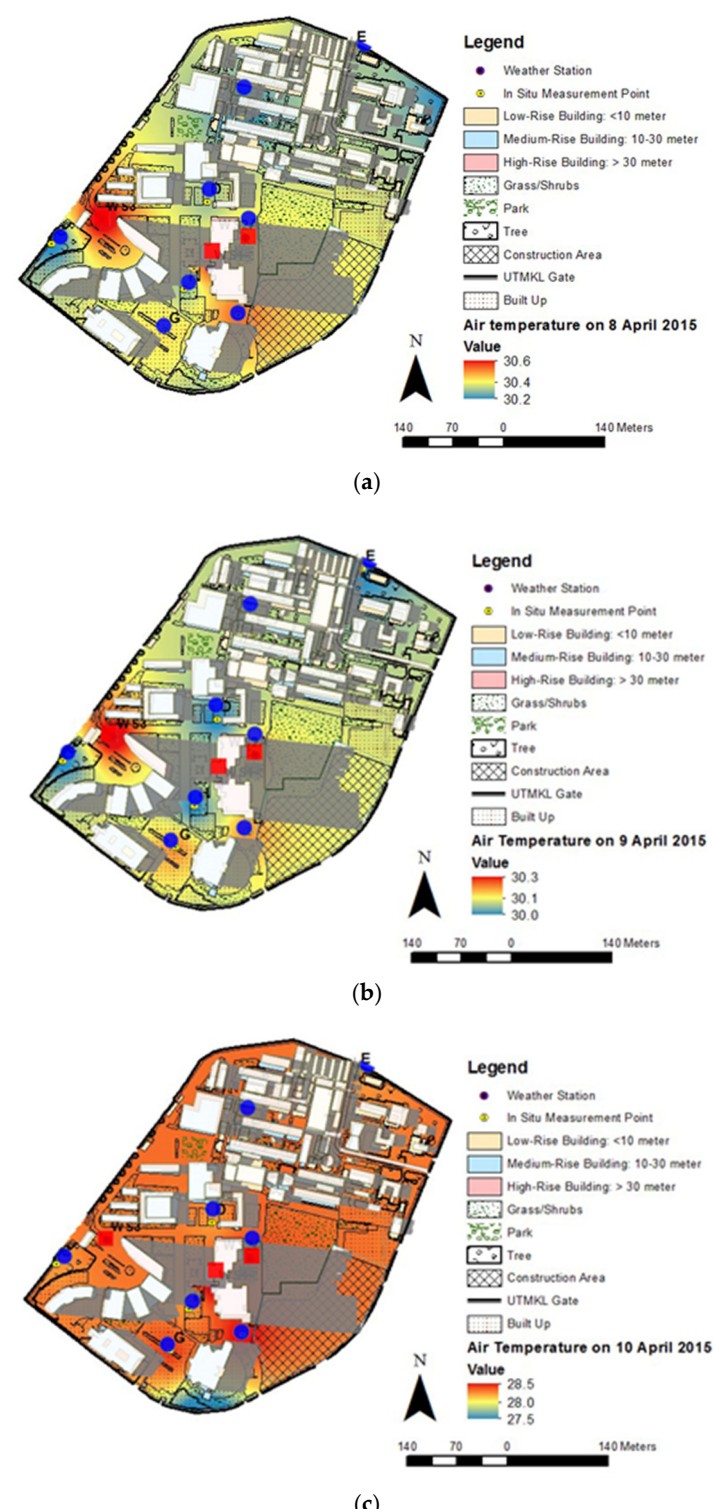

(**a**)

(**b**)

(**c**)

**Figure 10.** *Cont.*

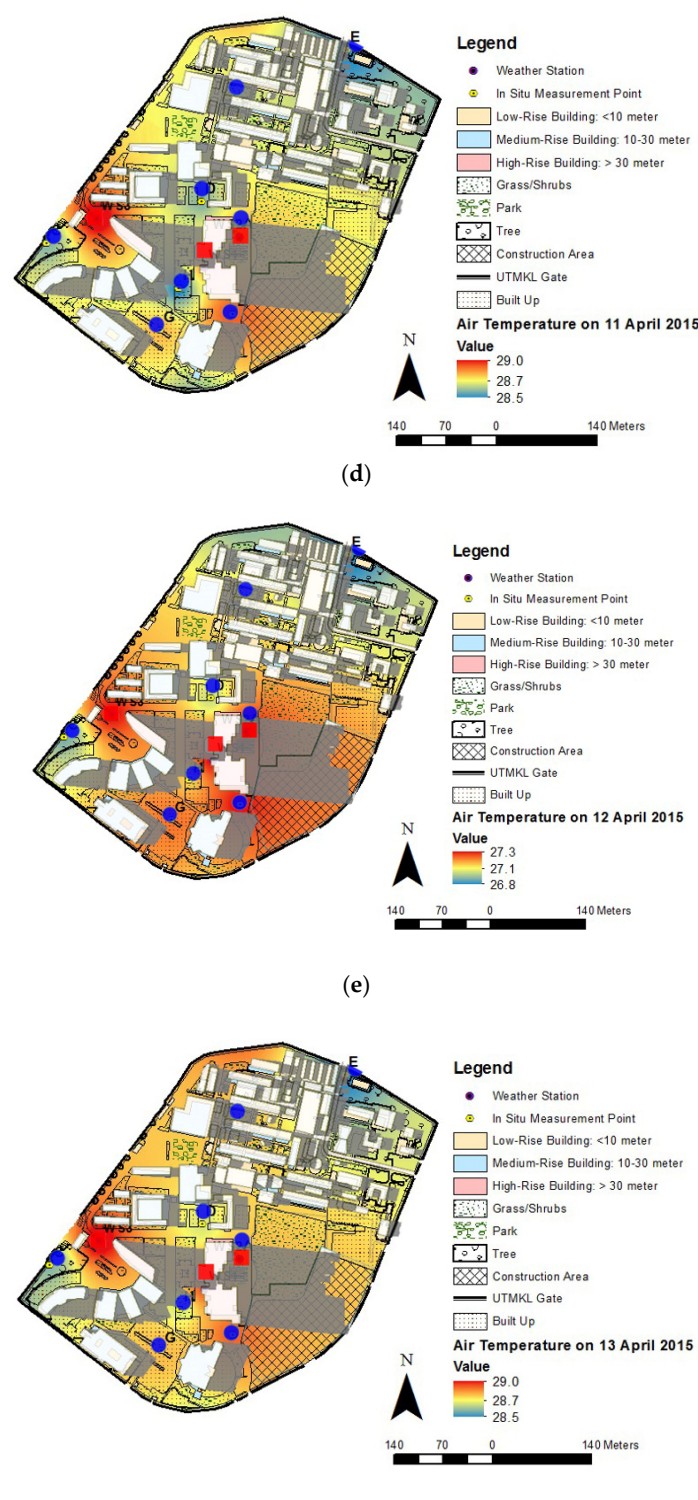

(**d**)

(**e**)

(**f**)

**Figure 10.** *Cont.*

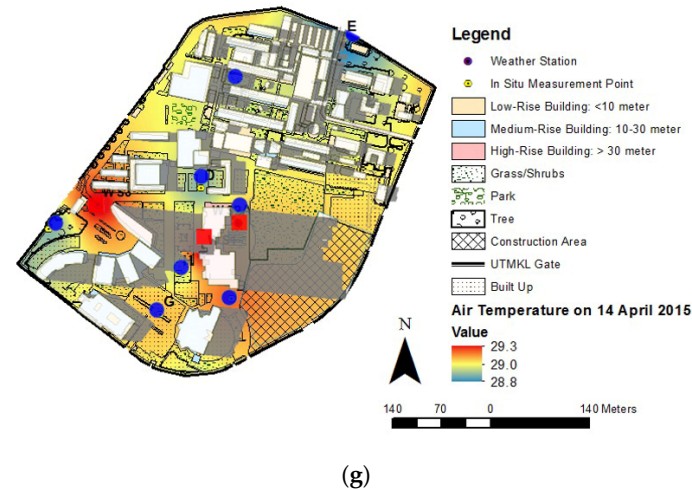

(**g**)

**Figure 10.** The mapping of UTMKL land cover features and the daily average air temperature interpolation (**a**) 8th April (**b**) 9th April (**c**) 10th April (**d**) 11th April (**e**) 12th April (**f**) 13th April (**g**) 14th April 2015.

The in-situ daily average air temperature indicated that point E (near the Noise and Vibration Lab) had a lower air temperature value, while points B (near the construction area), A (near the UTM KL field), and G (at the parking lot) had higher air temperatures compared to other points. Thus, the daily average air temperatures at points C (near the UTMKL main gate), D (near the tennis court), F (between blocks A and B), and H (between Razak Tower and MJIIT) are between the higher (points A, B and G) and lower (points E) average daily air temperatures.

The average air temperatures from 8th until 14th April 2015 at eight in-situ measurement points are shown in Table 4. Based on the table, the high area ratio of building to greenery (1:1.01) at point F resulted in the higher air temperature (28.9 °C). In addition, higher daily average air temperatures were observed at open spaces such as at points A and G. Both points displayed high SVF values (point A, 0.75; point G, 0.97), but low *H/W* ratios (point A, 0.49; point G, 0.18). The highest daily air temperature was observed at point B where the greenery ratio was the lowest (0.08). Moreover, the location of point B was surrounded with buildings (SVF = 0.79; *H/W* = 0.89), near the construction area, and exposed to solar radiation for the longest hours.

Meanwhile, low daily average air temperatures were recorded at the in-situ measurement point E as the area had a low building percentage (19%) and a high greenery percentage (22%). It also had a low SVF value (0.62) due to obstruction by buildings and trees. This resulted in the lowest daily average air temperature at point E. The low SVF value was also observed at the in-situ measurement points C and D where the daily average air temperatures were between the highest and the lowest air temperatures.

The daily average air temperature at point H which was located between two high-rise buildings (SVF = 0.83; *H/W* = 1.26) was neither high nor low because the location was shaded with building shadows and balanced with the presence of buildings and greenery at the ratio of 0.1 to 0.25. Overall, the areas with high building ratios, low greenery ratios, low SVF values, and low *H/W* ratios were observed to have higher daily average air temperatures. The GIS interpolation mapping displays the variation of the daily average air temperatures throughout UTMKL. Therefore, the spatial map presents all the essential information of land cover features, air temperature interpolation, SVF, and *H/W* ratio to critically evaluate the relationship of the parameters involved.

## 4. Conclusions

The paper draws two major conclusions based on the research purpose of this study. First, the variation in building morphology generates different shadow patterns that directly influence the outdoor air temperature. The formation of shading by the building that took place early in the

morning and in the evening was significant. The building ratio, SVF, and *H/W* ratio played an important role in the occurrence of building-induced shadow pattern. The average air temperature was found higher with a high building ratio, low greenery ratio, higher SVF value, and low *H/W*. Meanwhile, areas with low SVF values had lower average outdoor air temperatures. Essentially, a larger area and a longer period of building shading effectively improved the microclimate of a shaded area. It can be observed that with the application of effective building morphology ratio will be beneficial for the planning of campuses in improving the microclimate and outdoor thermal comfort.

Secondly, the findings suggested that the presence of greenery contributed to enhancing microclimate improvement. This parameter cannot be neglected due to its ability to further reduce the SVF value. Simultaneously, a combination of building-induced shading and greenery was shown to reduce the ground and air temperatures. It can be seen when the area was shaded independently by building shading caused higher average air temperature. However, for areas that were obstructed by buildings and trees, the air temperature was reduced due to low resulting SVF. Therefore, building morphology and greenery are the two parameters that should be considered in campus planning. The findings of this study are expected to benefit the future planning of the university campus. Besides, the importance of building morphology in creating appropriate shading patterns is vital knowledge in creating a better and comfortable campus environment. In addition, future studies may require a variation in building typology and green density to further explore the impact of these parameters on the microclimate of the campus area.

**Author Contributions:** Conceptualization, S.A.Z.; data curation, S.A.Z. and S.W.S.; formal analysis, S.A.Z. and S.W.S.; funding acquisition, S.A.Z., F.Y., M.Z.H., M.Y.M.D., and, M.I.A.; investigation, S.A.Z. and S.W.S.; methodology, S.A.Z.; project administration, S.A.Z.; resources, S.A.Z.; software, S.A.Z.; supervision, S.A.Z.; validation, S.A.Z. and S.W.S.; visualization, S.A.Z., and, S.W.S.; writing—original draft, S.A.Z., S.W.S., and M.F.S.; writing—review and editing, S.A.Z., and M.F.S. All authors have read and agreed to the published version of the manuscript.

**Funding:** This work was supported by the Ministry of Education (MOE) through the Fundamental Research Grant Scheme (RGS/1/2019/TK07/UTM/02/5), Universiti Teknologi Malaysia (UTM) under Industrial-International Incentive Grant (Vot 01M89), UTM Encouragement Research (Vot 18J24), and Research University Grant (1001/PTEKIND/8014124) from Universiti Sains Malaysia.

**Conflicts of Interest:** The authors declare no conflict of interest. The funders had no role in the design of the study; in the collection, analyses, or interpretation of data; in the writing of the manuscript, or in the decision to publish the results.

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
