# Peer review of "Assessment of Outdoor Air Temperature with Different Shaded Area within an Urban University Campus in Hot-Humid Climate"

_sustainability, doi:10.3390/su12145741_

Round 1

Reviewer 1 Report

The manuscript reports the assessment of outdoor air temperatures across a campus environment in Malaysia for seven days. Based on shadows from buildings and different urban morphological parameters, the study presents patterns that can increase or decrease the air temperature.

Despite presenting an interesting approach to investigate the impacts of these parameters on the microclimate of the campus area, there are essential points the authors should address:

1. Introduction:

- The study presents a good literature review but it is all concentrated in the Introduction. There are different paragraphs addressing the same subject, which makes the text a little repetitive. Some of the literature would fit better in the Discussion section, where the authors could use them to support their findings.

- I suggest the authors improve the description of the urban morphological parameters used in the study, to ensure that readers understand the relationship between the range of values and the increase/decrease in air temperature. That could be summarized in a table in section 2, for example, where the methods are described.  

2. Methods:

- The authors should include a table (or describe in the text) the accuracy/precision of the sensors used to measure air temperature. It is also important to describe the weather systems/ synoptic conditions from 8 to 14 April 2015, because those characteristics strongly influence the spatial distribution and intensity of temperature.

- Fig.1. Maybe add transparency to the colors so it is easier to identify the location of measurement points and weather stations.

- Table 2. If it is possible, add a column with pictures from the real canyons.

- Table 3. Increase the resolution and size of pictures so readers can see them better.

3. Results and Discussion:

- To assess the influence of shaded areas on air temperature, I would recommend analyzing their diurnal pattern together. The authors only present the shadows at one specific time over the seven days (e.g. Fig. 6. Daily building shadows in UTMKL at 12.00 pm from 8 to 14 April), and hourly patterns for one day (Fig. 7. Hourly building shadows on 8 April 2015).

- The authors analyze Fig. 7 and say "The building shadows lowered the air temperature at the shaded area. However, point B had a high average air temperature because the point was only covered by building shadows at 7.20 pm and exposed to the sunshine in the remaining hours.", but that information is not illustrated. The analyzes should be improved by plotting the measured temperatures (e.g. isotherms) within the hourly building shadow maps (overlay) - similar to the figures in Table 4.        
                                                                                                              - When discussing the relationship between outdoor air temperature and land cover, it would be more representative to include the temperatures from all days of measurement, instead of using just the daily air temperature measurement on April 11.

- Fig. 8 is a good illustration of land cover features and the average daily air temperature, but it only represents one day (April 11). To show a more representative pattern, I suggest to elaborate a map for each day of measurement or to show the spatial distribution of air temperature with the land cover features for different hours of the day. That would enable the authors to identify relationships between the land cover and the increase/decrease of temperature during the day and nighttime.

4. Conclusion:

- The study aims to demonstrate how the variation in building morphology generates shadow patterns that influence outdoor air temperatures. However, the discussion of results is based on one episode (time/day) and does not fully support the conclusions.       

- The authors should develop their analyzes and use the literature review to support their findings.

Author Response

Thank you for your comments. The details response as attached.

Reviewer 2 Report

An evaluation of the article "Assessment of outdoor air temperature with different shaded area within an urban university campus in hot-humid climate" was carried out.

The following are some points and notes:

The authors investigated the thermal variation of air in a shaded area built on a Malaysian university campus.

Initially, the article presents a great historical evolution with a wide variety of authors and the state of the art on the studies of heat islands, urban gorges, canyon geometry. It presents recent articles from the international literature, highlighting the novelties of the urban climate, but it also highlights classic studies such as Oke, Johnson and others.

Figure 1 deserves some adjustments:
The source of the caption can be larger;
The black color of the letters on the map can be changed to yellow to facilitate contrast and identification of observation points;
Insert a color satellite image on the right side of the map;
Insert a map of geographic characterization - map of Malaysia, followed by the city of Kuala Lumpur and location of the university campus in the city.

The descriptive source in Figure 2 is outside the journal's norms. It deserves a revision throughout the text.

In lines 218 and 2019 "The building shadows were created by setting the [visualizations of sun path and sun orientation]", can be changed to "apparent movement of the Sun".

Can the figures in Table 3 be larger? It is difficult to visualize it.

Figure 5 shows the temperature of the outside air. However, it is suggested that the authors discuss the causes of the temperature reduction on April 12, 2015. Is it possible that there was a change in the atmospheric weather with cloudiness and precipitation? A synoptic description of the weather is necessary.

It is also suggested that the authors provide a description of the climatic characteristics of Kuala Lumpur. This is not clear from the text. This enriches the characterization and discussion of the work.

The study presents an important contribution to university campus planning and possibilities for adapting the built environment to the thermal comfort of university professors, researchers, technicians and students.

Author Response

(The authors gave the same response as above.)

Reviewer 3 Report

Overall, this paper is very interesting with the main focus on the relationships between building-induced shadow and outdoor air temperature. This paper is important to present the evidence. The experimental has been well designed but has not been well presented. I believe authors have considered such. I suggest authors do some necessary revisions. 

  1. Some description is not proper, I have marked some of them in the abstract ( see attachment). However, authors should further double check the remaining. 
  2. The introduction should be rewritten. (1) It is too long (2) authors have not directly touch the key thing of the relationship between urban morphology and outdoor thermal environment or the UHI. Authors must present this story line clearly. Some comments are given in the attachment.
  3. The research gap this paper has not been well presented, and the research aim/objective of this paper has not been well defined (main focus is about the building-induced shadow, while authors' presentation in the abstract and conclusion has also toughed the vegetation). authors should reconsidered the objectives (at the end of the introduction).
  4. The experimental design has not been well presented, what is the weather condition? cloud condition? what is the information of the case study area? what is the information of the weather station?....
  5. The results in the Figure 5 should be presented by the bar chart with error bar.
  6. The results in Fig.6 and Fig.7 should be linked with the temperature. 

Please see the attachment about the minor comments.

I believe authors can revise this paper well. Good luck.

Author Response

(The authors gave the same response as above.)

Round 2

Reviewer 1 Report

The paper shows good improvement over the first submission. I would only recommend the authors to consider the suggestions:
(a) to include the atmospheric conditions from 8 to 14 April 2015, because those characteristics strongly influence the spatial distribution and intensity of temperature; and (b) improve the conclusion. Those topics were not presented in the cover letter.

Reviewer 3 Report

well done

Author Response

Thank you. There are no comments from reviewer.